# Therapeutic Potential of Various Intermittent Fasting Regimens in Alleviating Type 2 Diabetes Mellitus and Prediabetes: A Narrative Review

**DOI:** 10.3390/nu16162692

**Published:** 2024-08-14

**Authors:** Sthembiso Msane, Andile Khathi, Aubrey Sosibo

**Affiliations:** Department of Human Physiology, School of Laboratory Medicine and Medical Sciences, College of Health Sciences, University of KwaZulu-Natal, Durban 4041, South Africa; 219023719@stu.ukzn.ac.za

**Keywords:** intermittent fasting, type 2 diabetes mellitus, prediabetes, HbA1c, glucose tolerance

## Abstract

Intermittent fasting has drawn significant interest in the clinical research community due to its potential to address metabolic complications such as obesity and type 2 diabetes mellitus. Various intermittent fasting regimens include alternate-day fasting (24 h of fasting followed by 24 h of eating), time-restricted fasting (fasting for 14 h and eating within a 10 h window), and the 5:2 diet (fasting for two days and eating normally for the other five days). Intermittent fasting is associated with a reduced risk of type 2 diabetes mellitus-related complications and can slow their progression. The increasing global prevalence of type 2 diabetes mellitus highlights the importance of early management. Since prediabetes is a precursor to type 2 diabetes mellitus, understanding its progression is essential. However, the long-term effects of intermittent fasting on prediabetes are not yet well understood. Therefore, this review aims to comprehensively compile existing knowledge on the therapeutic effects of intermittent fasting in managing type 2 diabetes mellitus and prediabetes.

## 1. Introduction

Intermittent fasting (IF) is a broad term describing several eating regimens in which individuals alternate long periods of normal calorie intake with periods of minimal or no energy intake [1]. In 1935, McCay elucidated the correlation between calorie restriction and lifespan or longevity [2]. Since then, studies have extensively investigated calorie restriction, evolving into the practice of intermittent fasting [1,3,4,5]. Research findings have highlighted the use of intermittent fasting and its efficacy in metabolic-related disorders [1,3,4,5,6]. Several intermittent fasting protocols have been recognized for their capacity to mitigate metabolic disorders [7,8,9,10,11,12,13,14,15].

The IF protocols include dietary approaches that involve alternating eating periods with either fasting by restricting calorie intake or zero calorie intake during the fasting period [6]. The timing of fasting and feeding periods varies among different IF protocols, such as the 5:2 diet, alternate-day fasting, and time-restricted feeding [16]. The 5:2 fasting diet is a dietary regimen where individuals eat without restrictions for five days, followed by two days per week during which they consume a very-low-calorie diet (fewer than 800 calories per day) [17]. Alternate day fasting (ADF) involves alternating between a 24 h fasting period, during which individuals consume less than 25% of their usual energy needs, and a 24 h eating period, where they can eat normally [15]. Time-restricted feeding (TRF) is an IF protocol with a specified time of prolonged fasting practiced by adhering to 16 h of abstinence from food and 8 h of food intake within 24 h [6]. IF has gained popularity in body weight management and alleviating metabolic-related disorders [18,19]. Therefore, this review aims to synthesize existing knowledge on the therapeutic effects of intermittent fasting on the management of T2DM and prediabetes, providing a critical overview of its current state of understanding. This review will begin by examining the impact of various intermittent fasting protocols on general metabolic issues, particularly those associated with type 2 diabetes (T2DM). Given that T2DM is often preceded by prediabetes, both metabolic states will be discussed. The effects and limitations of conventional management strategies for T2DM and prediabetes will be briefly described. Subsequently, this review will explore the impact of all intermittent fasting regimens on T2DM and prediabetes. Then, future recommendations and conclusions about the possible use of IF regimens in diabetes management will be made.

## 2. Effects of IF on Metabolic Complications

Obesity has emerged as a notable issue. For example, a study reported an obesity-related death rate of 62,000 out of every 100,000 individuals in the population [20]. Epidemiological studies utilize Body Mass Index (BMI) as a tool to identify individuals who are either obese or overweight [21,22]. Obesity has been correlated with the development of several physiological disorders, including type 2 diabetes, inflammation, cardiovascular disease, hypertension, dyslipidemia, non-alcohol fatty liver disease, and insulin resistance [21,22]. Energy imbalance, leading to excess body fat (≥20%), defines obesity [23,24,25]. Therapeutic interventions such as fasting and the use of antidiabetic medications have been linked to substantial weight loss, suggesting improvements in clinical factors associated with metabolic complications [26,27,28,29].

The fasting regimens mentioned have demonstrated efficacy in eliciting favorable metabolic alterations. The changes include improved glucose control, reduced glycogen storage, release of fatty acids and ketones, decreased levels of leptin, and increased levels of adiponectin [1,30,31,32]. In overweight or obese adults, IF has reported a decrease in BMI, body weight, waist circumference, and fat mass [3,17,33,34,35]. Interestingly, a study conducted in obese middle-aged female Wistar rats found that ADF and TRF did not lead to reductions in blood lipid profiles, adiposity, or insulin resistance. Instead, these dietary interventions increased inflammatory biomarkers, potentially elevating the risk of obesity-associated comorbidities [36]. Other studies reported different outcomes of insulin resistance, blood lipids, adiposity, inflammatory markers, and glycemic control upon IF adherence [37,38].

Insulin resistance (IR) occurs when the main target tissues for insulin action in glucose metabolism do not respond to insulin as they should due to chronic energy surpluses [39]. Therefore, weight loss is essential for regulating disordered glucose and lipid metabolism, notably insulin resistance and hyperinsulinemia caused by central obesity [30,31,40]. Furthermore, insulin resistance has been implicated in T2DM [41,42]. T2DM is a chronic hyperglycaemic condition triggered by a preceding loss of β-cell insulin secretion and insulin resistance [43]. The onset of a metabolic switch brought on by fasting is due to the negative energy balance caused by the depletion of glycogen stores and metabolized fatty acids.

The metabolic switch from using glucose to fatty acid-derived ketones represents a gradual change in the metabolism from lipid/cholesterol synthesis and fat storage to fat mobilization through fatty acid oxidation and fatty acid-derived ketones. As a result, the metabolic switch aids in maintaining muscle mass and its function, which promotes weight loss [16]. During fasting, ketones continuously increase while glucose decreases, and this is inverse to the postprandial state, where glucose levels increase while ketones diminish [44]. The outcomes of the metabolic transition observed during fasting may ameliorate insulin sensitivity and glycaemic regulation detected in persons with non-insulin-dependent diabetes.

## 3. Type 2 Diabetes Mellitus

Diabetes continues to be a significant contributor to mortality and morbidity rates globally [45]. It is expected that 578 million people will have diabetes by 2030, increasing by 51% (700 million) by 2045 from 463 million in 2019 [45,46]. Approximately 90% to 95% of all diabetes diagnoses are classified as type 2 diabetes mellitus [47]. According to both the American Diabetes Association (ADA) and World Health Organisation (WHO), diabetes can be diagnosed if a person has a fasting plasma glucose level of ≥126 mg/dL (7.0 mmol/L) after fasting for at least 8 h and a plasma glucose level of ≥200 mg/dL (11.1 mmol/L) during a 75 g oral glucose tolerance test (OGTT) [48]. The ADA also uses a glycated hemoglobin (HbA1c) level of at least 6.5% (48 mmol/mol Hb) to diagnose diabetes [49,50]. Both the ADA and the WHO approve diagnosing glucose in plasma, although it can also be measured in serum and whole blood [51,52].

Multiple organs are involved in the pathophysiology of T2DM. Disruption in the pathways connecting the endocrine pancreas, liver, skeletal muscles, and adipose tissues can disrupt glucose regulation, leading to the onset of T2DM, as illustrated in Figure 1. The primary causes of the disruption include chronic inflammatory markers (e.g., Interleukin-6 and C-reactive protein) and overnutrition, especially through diets high in carbohydrates and saturated fats [53]. They disrupt the insulin signaling pathway in muscle, liver, and adipose tissues, resulting in hyperglycemia. Progressive hyperglycemia further damages the β-cells of pancreatic islets by inducing oxidative stress within the β-cells of pancreatic islets [54].

## 4. Conventional Management of T2DM

### 4.1. Insulin Therapy

Insulin therapy serves as an injectable medication for diabetes mellitus [55]. The use of exogenous insulin is primarily employed to regulate blood glucose levels and alleviate symptoms of T2DM by replenishing or complementing the body’s natural insulin production from the pancreas [55,56]. Insulin therapy directly activates the insulin receptor, leading to increased glucose uptake, decreased production of glucose by the liver, and decreased breakdown of fats [57]. Hence, insulin therapy is positively associated with a decrease in fasting glucose levels, glucose tolerance, and HbA1c [58,59,60]. Nevertheless, diminishing fat breakdown may lead to the buildup of triglycerides in both the bloodstream and fatty tissues, resulting in an elevated likelihood of gaining weight [61]. The weight gain mechanism in insulin-treated T2DM involves factors such as hypoglycemia-associated snacking, inhibited glucose excretion (glycosuria), and decreased metabolic rates [62].

### 4.2. Glucagon-like Peptide-1 Receptor Agonists

Glucagon-like peptide-1 (GLP-1) is a hormone primarily located in the gastrointestinal tract, secreted in response to the ingestion of nutrients (carbohydrates and fats) [63]. GLP-1Ra functions by controlling elevated blood glucose levels after meals by increasing the release of insulin from the beta cells [64]. These drugs have been associated with decreased FG, HbA1c, and GT [65,66,67]. Additional benefits encompass suppressing glucagon release, delaying stomach emptying, promoting insulin release, and reducing appetite [63,64,68]. The disadvantages of using GLP-1Ra have been associated not only with gastrointestinal effects but also with gallbladder diseases, attributed to reduced gallbladder refilling [69,70].

### 4.3. Dipeptidyl Peptidase-4 Inhibitors

Dipeptidyl peptidase-4 inhibitors (DPP4i) function by inhibiting the activity of DPP-4 [71]. DPP4i enhances insulin secretion from pancreatic beta cells in a glucose-dependent manner through the action of GLP-1 while concurrently decreasing glucagon release from alpha cells [68,72]. DPP-4 inhibitors have been demonstrated to effectively lower FG, postprandial glucose, and HbA1c levels while maintaining a low risk of hypoglycemia [72,73]. Nonetheless, there has been a noted rise in the occurrence of acute pancreatitis linked to their utilization [71,74].

### 4.4. Sodium-Glucose Co-Transport 2 Inhibitors

Sodium-glucose co-transport (SGLT)-2 is a kidney transporter responsible for the reabsorption of glucose from the renal filtrate, thus hindering the excretion of glucose through urine [75]. SGLT2 inhibitors represent a newer class of antihyperglycemic medications that function independently of insulin, providing effects beyond simply lowering glucose levels. These drugs promote urinary glucose excretion and natriuresis by inhibiting the reabsorption of glucose and sodium in the proximal tubule of the kidney [75]. This class of antidiabetic drugs has been revealed to reduce HbA1c levels (~0.51–1.01%), FG, and postprandial glucose [76,77,78]. The disadvantages related to the administration of SGLT2i include increased risk of urinary tract infection, genital infection, and lower limb amputation [79,80].

### 4.5. Biguanide (Metformin)

Metformin, a commonly prescribed antidiabetic drug, is widely acknowledged as a biguanide with properties that enhance insulin sensitivity [81]. Metformin improves glucose utilization and sensitivity to insulin in tissues outside the liver [82]. At therapeutic doses, metformin utilizes multiple mechanisms to reduce blood glucose levels [83]. This antidiabetic drug has been reported to improve GT, HbA1c, and FG [83]. While the liver is the main target organ for metformin action, there is also evidence suggesting involvement of the intestines [84].

Metformin’s effects in the gastrointestinal tract encompass increased intestinal absorption and lactate generation, elevated concentrations of GLP-1, and modification of bile acid pools, consequently impacting the microbiome’s composition [84]. However, the use of metformin has been associated with the occurrence of vitamin B12 deficiency, which may contribute to the manifestation of diabetic neuropathy symptoms [81,85]. Furthermore, changes in gut flora, alterations in gut motility, competitive inhibition of absorption, and impairment of calcium-dependent membrane actions in the terminal ileum have been proposed as mechanisms contributing to the development of vitamin B12 deficiency associated with metformin use [85,86]. Regular monitoring of vitamin B12 levels and appropriate supplementation may be necessary for individuals on long-term metformin therapy to address this potential concern. Different types of antidiabetic drugs and their mode of action are shown in Table 1.

## 5. Lifestyle Intervention

Lifestyle intervention is widely recognized for its effectiveness in reducing the risks associated with T2DM [87]. Lifestyle intervention has been linked to reduced occurrences of T2DM and lower incidences of cardiovascular events, microvascular complications, cardiovascular mortality, and all-cause mortality, leading to increased life expectancy in patients with IGT [87]. The application of lifestyle intervention has also been reported as cost-effective for patients who adhere to it [88]. A DPP 10-year follow-up diabetes study showed that at 2.8 years, there was a 58% reduction in the incidence of diabetes among high-risk adults with lifestyle intervention, which was superior to the 31% reduction observed with metformin [89]. In the initial year visit, a mean weight loss of 7 kg was observed [89]. This confirms the superiority of lifestyle intervention over the established first-line drug, metformin. Intervention strategies encompass physical activity, exercise, and dietary plans [90].

### 5.1. Dietary Intervention

Dietary intervention encompasses the banting diet, the ketogenic diet, and the Mediterranean diet [91,92,93]. The Banting diet is characterized by high protein intake, whereas the ketogenic diet focuses on low carbohydrates, high fat, and adequate protein, and the Mediterranean diet emphasizes a higher consumption of vegetables [91,93,94]. To achieve long-term weight loss, factors such as meal timing and macronutrient composition must counteract compensatory changes in hunger, cravings, and ghrelin suppression mechanisms. These factors can serve as a boost for weight gain after a previous loss [95]. However, dietary intervention has been linked with positive effects on FG, GT, and HbA1c [96,97,98]. Abstinence from food or fasting entails the breakdown of lipids, carbohydrates, and proteins to regulate plasma glucose within the normal range. Progressive accumulation of fats in the pancreas and liver may lead to dysfunction of beta cells, resulting in hyperglycemia. This condition can be reversed by reducing fats in the liver and pancreas [5].

### 5.2. Increased Physical Activity

Physical activity has demonstrated antidiabetic effects in individuals with T2DM [99]. Physical activity consists of body movements driven by the contraction of skeletal muscles, resulting in increased energy expenditure [99]. Increased physical activity, such as exercise interventions, has been implemented to alleviate hyperglycemia [100]. Research has shown that low- and moderate-intensity exercise can lower FG, GT, and HbA1c levels [101,102]. Qualitative research revealed that obstacles to physical activity can include health issues (such as breathing problems), difficulties with time and lifestyle management (such as lack of time and motivation), and various environmental, social, and cultural factors [103].

## 6. Effect of Intermittent Fasting on T2DM

### 6.1. Alternate Day Fasting 

Many investigations have been carried out to assess the safety and tolerability of alternate-day fasting regimens, showing promising clinical outcomes related to T2DM [104,105,106]. Research findings suggest that alternate-day fasting can serve as an alternative approach to continuous CR, with superior effects observed in the retention of lean mass [105,107]. The utilization of alternate-day fasting resulted in a significant decrease in total cholesterol and serum triglycerides [108]. Another study, supported by evidence, demonstrated that adherence to alternate-day fasting can positively impact glucose tolerance within 3 weeks via heightened expression of the SIRT1 gene [109]. Despite a notable reduction in total intra-abdominal fat mass, the alternate-day fasting group reported a failure to alleviate diet-induced muscle insulin resistance caused by a high-fat diet [110].

The potential cause may be attributed to a decrease in the expression of GLUT-4 protein in both high-fat ad libitum (HF-AL) and high-fat alternate-day fasting (HF-ADF) rats compared to the Chow group [110]. Conversely, another study showed a positive impact on glycemic control in genetically obese mice undergoing alternate-day fasting despite the absence of significant weight loss [111]. In mice, fructose-induced resistance was alleviated by a 100% restriction on chow food but allowing ad libitum access to a fructose drink during the fasting days of alternate day fasting. This group exhibited significant improvements in insulin sensitivity compared to the control group [112]. A study revealed a direct correlation between alterations in body weight and improvements in glycemic control, insulin sensitivity, and insulin secretion in obese males with and without T2DM [113]. Alternate-day fasting produces superior outcomes, specifically a decrease in fasting insulin levels and insulin resistance, compared to continuous CR in individuals with insulin resistance [34].

Alternate fasting has also been associated with adverse effects, including hunger, impaired cognitive function, and irritability, which may diminish within a month of adherence [114,115]. On the contrary, a six-month study found that combining ADF with a low-carbohydrate diet did not result in changes in appetite [116]. This suggests that various alternate-day fasting protocols may lead to diverse outcomes upon adherence. Therefore, further research is necessary to assess the impact of different alternate-day fasting approaches on the body’s physiological functions.

### 6.2. The 5:2 Fasting Diet

The 5:2 diet regimen may serve as an alternative approach to continuous CR, exhibiting reported comparable efficacy in weight management and glycemic control [117]. A (600 kcal)/day diet has been demonstrated to yield significant improvements in beta cell function and hepatic insulin sensitivity, potentially leading to a reversal of T2DM [118]. A 12-week study comparing consecutive versus non-consecutive fasting days utilizing the 5:2 diet regimen demonstrated significant reductions in weight and glycemic levels among individuals with T2DM [119]. Dietary restriction of energy intake was associated with substantial improvements in various markers, including reductions in HbA1c levels, improved results in OGTT, decreased pancreatic and liver triacylglycerol stores, and lowered FG levels [118]. However, cautious measures may be necessary for continuous VLCD regimens, particularly in managing oral hypoglycemic agents to prevent hypoglycemia [120]. Additionally, VLCDs can pose long-term risks of complications such as micronutrient deficiencies [121]. The 5:2 diet has been associated with significantly lower compliance rates. It has been indicated that this diet often results in significant overcompensation during non-fasting days [4].

### 6.3. Time-Restricted Feeding 

TRF has been documented to exhibit a high adherence rate among participants [122]. This regimen has been shown to positively influence fasting glucose, glucose tolerance, and HbA1c levels in T2DM [123]. The glycemic impacts can be achieved through the Circadian Timing System [124]. Several studies have proven that aligning with the circadian timing system plays a role in positive outcomes upon TRF adherence [125,126,127,128]. Research involving both humans and animals suggests that early TRF is favored as the superior regimen over late TRF [125,128]. Early TRF has been shown to enhance insulin sensitivity, promote weight loss and fat oxidation, and help manage glycemic levels [128]. Different types of intermittent fasting regimens are shown in Table 2.

## 7. Prediabetes

Despite the possibility of correcting prediabetes to normal glucose regulation, it nonetheless imposes a strain [134]. Prediabetes can be characterized by elevated blood glucose levels that do not meet the diagnostic criteria for diabetes [135]. Prediabetes has an asymptomatic characteristic, and this makes it hard to diagnose [136]. The prevalence of prediabetes is increasing year after year, with 5% to 10% of prediabetic people advancing to fatal T2DM and its related complications [31,136,137]. The diagnostic criteria of prediabetes include FG, IGT, and HbA1c. However, the WHO does not recognize HbA1c as a diagnostic criterion for prediabetes. It has been shown that the ADA identifies individuals with an IFG of 5.6–6.9 mmol/L, an IGT of 7.8–11.0 mmol/L, and a HbA1c of 5.7–6.4% as prediabetic [138]. The rising prevalence of prediabetes is causing significant concern. Recent research suggests that the global prevalence of prediabetes is expected to exceed 400 million individuals by 2045 [139].

Research has demonstrated that insulin resistance in adipose tissue contributes to the onset of hyperglycemia and associated complications [140,141,142,143]. Insulin resistance in adipose tissue stimulates the increased release of free fatty acids into the bloodstream, facilitating ectopic fat storage [141]. This process induces insulin resistance in the liver and skeletal muscles, culminating in metabolic issues such as elevated glycemic levels, abnormal lipid levels, hypertension, metabolic syndrome, and NAFLD [140,141,142]. Decreased physical fitness has been linked to increased levels of free fatty acids, reduced insulin clearance, diminished insulin sensitivity in muscles, slightly elevated triglycerides, and decreased levels of HDL cholesterol [142]. Additionally, research indicates that prediabetic individuals with insulin resistance face double the risk of cardiovascular disease compared to prediabetic individuals who do not have insulin resistance [144]. Prompt detection of prediabetes and its associated complications is vital for mitigating the risks it poses to individuals.

### HOMA-IR

The Homeostasis Model Assessment of Insulin Resistance (HOMA-IR) represents a diagnostic tool used in clinical settings to evaluate the resistance of insulin, calculated as HOMA-IR = (Fasting Insulin (mU/L) × Fasting Glucose (mmol/L))/22.5 [145]. Prediabetic individuals have insulin resistance in their hepatocytes, fatty tissues, or skeletal muscles [146].

## 8. Prediabetes Management

Irrespective of the current criteria employed for prediabetes diagnosis, the presence of IR, obesity, and either IFG, IGT, or both, still poses the risk of progressing to T2DM [21,22,144,147]. Various measures have been utilized to decrease prediabetes prevalence and progression to T2DM. Primary treatments for prediabetes include a combination of lifestyle changes such as weight loss and increased physical activity, as well as the use of medications such as metformin [148]. Lifestyle changes have demonstrated effectiveness in reducing the risk of developing T2DM, even with less intensive interventions [149].

### 8.1. Biguanides (Metformin)

Metformin is a medication used for managing both prediabetes and T2DM [83]. Interestingly, research indicates that metformin has beneficial effects on glucose measures such as FG, GT, and HbA1c levels [150,151]. However, using metformin alone is less effective than combining it with other antidiabetic medications or lifestyle changes [151,152]. Furthermore, metformin use is associated with adverse effects such as lactic acidosis, vomiting, and diarrhea [150].

### 8.2. Lifestyle Modification

A lack of physical activity and obesity significantly contribute to the advancement of T2DM [153]. Physical activity involves the body’s movement through the contraction of skeletal muscles, which leads to an elevation in energy expenditure [154]. Increased physical activity, such as exercise interventions, has been implemented to alleviate prediabetes [100]. Exercise interventions have improved IFG, IR, IGT, HbA1c levels, and weight loss. Exercise is associated with muscle insulin sensitivity [155]. An increase in insulin sensitivity is facilitated by the movement of several GLUT4 transporters to the cell membrane in response to a submaximal insulin stimulus [155], thereby promoting glucose tolerance and reducing glucose levels in the bloodstream [100].

However, short exercise interventions have been reported to fail to alter HDL-C levels [156]. A multivariate analysis found that the duration of exercise per session is a key predictor of changes in HDL cholesterol levels [157]. Additionally, the efficacy of exercise intervention in raising HDL-C levels has been associated with lower BMI or higher total cholesterol levels [157,158]. Variations in blood glucose levels could be affected by the type of physical activity engaged in, and specific exercise modalities might not be viable options for individuals who are overweight or obese [159]. Therefore, overweight or obese prediabetic individuals need to prioritize weight loss as a preliminary step to enhance their HDL-C levels.

Weight loss strategies often involve dietary interventions, which typically entail reducing calorie intake to manage body weight and address other clinical factors [160]. Very-low-calorie restriction has been linked to improvements in beta-cell function, leading to the restoration of the first phase of insulin secretion in prediabetic individuals [161]. Studies have reported a reduction in fasting glucose levels and HbA1c, weight loss, increased fasting insulin levels, and improved Homeostatic Model Assessment of Beta-cell Function (HOMA-β) [161,162]. Nevertheless, elevated fasting ghrelin levels have been correlated with an increased risk of weight regain following weight loss [163]. Interestingly, achieving positive metabolic outcomes through intermittent fasting may not necessarily require weight loss [164].

### 8.3. Intermittent Fasting

#### 8.3.1. Alternate-Day Fasting 

Recent research has emphasized the potential of IF as an alternative approach for addressing metabolic factors associated with prediabetes [52,104,164]. The literature has shown the efficacy and safety of ADF, the 5:2 fasting diet, and TRF in managing prediabetes [165,166]. Although Ingersen and colleagues reported a lack of significant changes in insulin sensitivity or secretion, other studies have suggested that ADF has the potential to decrease body weight, lower fasting insulin levels, improve IFG, reduce postprandial hyperglycemia, and decrease levels of HbA1c [34,113,167].

#### 8.3.2. The 5:2 Fasting Diet

The 5:2 fasting diet has demonstrated notable efficacy in diminishing body weight, improving insulin sensitivity, and lowering both IFG and HbA1c over a 12-week intervention period [168]. Despite achieving favorable results, the effectiveness of fasting methods in enhancing metabolic factors might fluctuate depending on the duration of fasting or the specific fasting strategies employed [52,166]. Fernanda et al. revealed that long-term intermittent feeding led to glucose intolerance while maintaining insulin receptor phosphorylation. It significantly increased insulin receptor nitration in both intra-abdominal adipose tissue and muscle, a modification linked to receptor inactivation [169]. Hence, this study aims to highlight the benefits of the IF regimen, which may serve as the standard alternative approach for prediabetes.

#### 8.3.3. Time-Restricted Feeding 

A commonly followed approach to TRF involves fasting for 16 h and consuming meals within an 8 h window each day (16/8) [170]. Alternatively, individuals may choose to fast for 14 h and consume meals within a 10 h window daily (14/10), or they may opt for a 20 h fasting period followed by a 4 h window for food consumption (20/4) [132,171]. Clinical studies have been drawn to these approaches due to their effects in regulating T2DM and its associated complications in short-term studies [12,172]. A systematic review revealed that the 16/8 and 14/10 fasting methods exhibit similar efficacy in weight loss [173].

Research has indicated the effectiveness of adhering to a 14/10 fasting regimen in controlling glycemic levels despite it not affecting insulin sensitivity in T2DM [37,174]. TRF has been demonstrated to increase insulin sensitivity, decrease blood glucose levels, reduce fasting insulin, reduce HbA1c levels, and enhance glucose tolerance by stimulating beta-cell responsiveness [12,37,132,164,172]. On the contrary, the literature has examined the effects of adhering to TRF either early or late in the day. The literature has observed that consuming meals late in the day leads to a suppression of resting energy expenditure, reduced fasting carbohydrate oxidation, and impaired glucose tolerance [175,176]. Meal timing plays a role in weight loss therapy, with delayed lunch consumption being associated with less weight loss compared to eating earlier, regardless of adhering to a hypocaloric diet [177]. Research consistently shows that consuming meals earlier in the day (8 a.m. to 7 p.m.) is notably more effective in reducing body weight, fasting glucose levels, and insulin resistance, enhancing insulin response, and decreasing ghrelin levels compared to eating later in the day (12 p.m. to 11 p.m.) [177,178,179]. Thus, early TRF has superior effects on metabolic factors compared to late TRF. Below is Table 3 summarizing the various types of IF regimens and their effects on prediabetes.

## 9. Conclusions

The available anti-diabetic medications have demonstrated the potential to improve T2DM and prediabetes. However, significant drawbacks have surfaced, creating an opportunity for alternative approaches. Numerous studies have examined the effects of intermittent fasting on T2DM. However, research on the relationship between intermittent fasting and prediabetes is still scarce. Intermittent fasting has been proven beneficial in short-term studies, highlighting its beneficial impacts on metabolic factors. These effects have been investigated in metabolic disorders such as obesity and T2DM. Nevertheless, the precise mechanisms through which IF modulates glycemic levels and its enduring influence on gene expression, such as GLUT 4 and IRS1, remain unclear. Additionally, the impact of intermittent fasting on glycemic markers and identifying an optimal IF regimen for prediabetes management is yet to be fully elucidated. Therefore, there is a need for comprehensive, long-term investigations to assess the role of IF in glycemic regulation and its associated gene expression, including examining key regulators such as GLUT 4 and IRS1.

## Figures and Tables

**Figure 1 nutrients-16-02692-f001:**
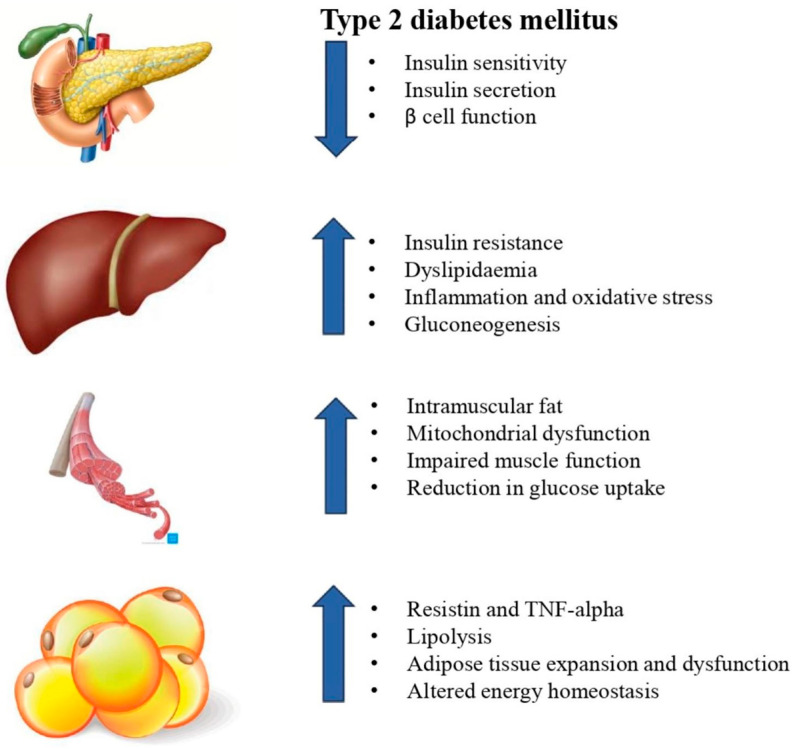
Illustrates the primary organs and molecules whose modifications contribute to the progression of type 2 diabetes mellitus.

**Table 1 nutrients-16-02692-t001:** Shows the different types of antidiabetic drugs and their mode of action, effects on glucose parameters, and shortfalls.

Types of Antidiabetic Drug(s)	Mode of Action	Effects on Glucose Parameters	Shortfall(s)
Insulin therapy	Direct glucose-lowering effect. Facilitation of glucose uptake by cells. Inhibition of hepatic glucose production. Promotion of glycogen synthesis.	Reduced FG, Reduced GT Reduced HbA1c	Weight gain
GLP-1RA	Slowing of gastric emptying Suppression of glucagon secretion Enhancement of glucose-dependent insulin secretion Improvement in Beta Cell Function Reduction of Appetite and Food Intake	Reduced FG, Reduced GT Reduced HbA1c	Gastrointestinal effects Gallbladder disease
DPP4i	Inhibition of DPP-4 enzyme Reduction in Blood Glucose Levels Prolongation of incretin hormone activity Weight management	Reduced FG Reduced GT Reduced HbA1c	Acute pancreatitis
SGL2i	Inhibition of SGLT2 in the kidneys Increased urinary glucose excretion. Reduction in blood glucose levels Caloric loss and weight reduction. Osmotic Diuresis.	Reduced FG, Reduced GT Reduced HbA1c	Urinary tract infection, Genital infection Lower limb amputation
Metformin	Enhanced peripheral glucose uptake. Inhibition of intestinal glucose transport. Improvement of lipid metabolism.	Reduced FG, Reduced GT Reduced HbA1c	Vitamin B12 deficiency Lactic acidosis

**Table 2 nutrients-16-02692-t002:** Summary of Different Types of Intermittent Fasting Regimens and Their Positive and Adverse Effects on Metabolic Disorders.

IF Regimen	Description	Positive Effects	Adverse Effects
5:2 fasting diet [4,119,129]	Involves a 5-day non-fasting period and a 2-day fasting period.	Improvements in weight, HbA1c, lipids, fasting glucose, and quality of life	Fasting elevated the occurrence of hypoglycemia even with reduced medication. Fasting may lead to over-compensation during non-fasting days.
Alternate day fasting [110,111,130]	Involves alternating between a 24 h fasting period, during which individuals consume less than 25% of their usual energy needs, and a 24 h eating period, where they can eat normally.	Decrease in body weight, waist circumference, systolic blood pressure, and fasting plasma glucose.	Fatigue, Headaches
16 h:8 h TRF [128,131]	Adhering to 16 h of abstinence from food and 8 h of food intake within 24 h	Heightens insulin sensitivity and fat oxidation and decreases body weight, fat profile, and inflammation	Hunger and irritability, Palpitations, dizziness, headache, abdominal pain, mood changes, vomiting, and hypoglycemia
14 h:10 h TRF [131,132,133]	Adhering to 14 h of abstinence from food and 10 h of food intake within 24 h	Reduced body weight, improved HbA1c, enhanced body composition, lowered blood pressure, and decreased lipids associated with cardiovascular disease.	Disrupted Social Eating Patterns, Palpitations, dizziness, headache, abdominal pain, mood changes, vomiting, and hypoglycemia

**Table 3 nutrients-16-02692-t003:** Summary of Different Types of Intermittent Fasting Regimens and Their Positive and Adverse Effects on Prediabetes.

IF Regimen	Description	Positive Effects	Adverse Effects
5:2 fasting diet [165,180].	Involves a 5-day non-fasting period and a 2-day fasting period.	Improvements in body weight, HbA1c, lipids, fasting glucose, and appetite score	Increased hunger
Alternate-day fasting [112,113,169]	Alternating 24 h fasting and 24 h eating period	Improved insulin sensitivity, decreased body weight, lower fasting insulin levels, improved IFG, reduced postprandial hyperglycemia, and decreased levels of HbA1c	Increased hunger, vomiting, redox imbalance, and glucose intolerance
Time-restricted feeding [175,176]	Fasting and consuming meals within a limited timeframe of 24 h.	Reduced FG, HbA1c, and postprandial glucose	Dizziness, hunger, nausea

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
