# Peer review of "Therapeutic Potential of Various Intermittent Fasting Regimens in Alleviating Type 2 Diabetes Mellitus and Prediabetes: A Narrative Review"

_nutrients, 2024, doi:10.3390/nu16162692_

Round 1
Reviewer 1 Report
Comments and Suggestions for Authors
This review aims to comprehensively compile existing knowledge on the therapeutic effects of intermittent fasting in managing type 2 diabetes mellitus and prediabetes. This point is valuable.
1) However, the logical structure of the article is chaotic. Please adjust the logical structure.
2) The section of overview of DM is too long, but the section of effects of IF on DM is too short. Please revise these parts.
3) Moreover, the article did not focus on the important issues. Please include T2DM only.
4) And this review is not a comprehensive review. This is a narrative review.
Comments on the Quality of English Language
English language of the article needs minor revisions
Reviewer 2 Report
Comments and Suggestions for Authors
The work entitled “Therapeutic Potential of Various Intermittent Fasting Regimens in Alleviating Type 2 Diabetes Mellitus and Prediabetes: A Comprehensive Review.” attempted to comprehensively compile existing knowledge on the therapeutic effects of intermittent fasting in managing type 2 diabetes mellitus and prediabetes. The main content of the manuscript summarizes the basic knowledge about type 2 diabetes. However, information related to Intermittent Fasting Regimens is deficient. Moreover, no table or figure related to the topical subject is presented in the manuscript.
The language of the manuscript could be improved for greater clarity and accuracy.
Reviewer 3 Report
Comments and Suggestions for Authors
In this literature review (Nutrients-3083304), Msane et al., have tried to address a very complex and socio-economically significant issue of the impact and management of T2DM through intermittent fasting (IF) regimen. Given the global pandemic of obesity, diabetes and associated metabolic syndrome and paucity of treatment options, IFs represent a very promising approach. The review is well written and will be appealing to both scientists and clinicians. Below are few suggestions/comments to improve it further.
Comments:
1. The review will greatly improve, with a small section on the molecular basis of T2DM and a Figure showing key organs and molecules whose alterations drive the disease. The authors could also indicate how and where the available therapeutics for T2DM work.
2. The authors should add a table recapitulating the various studies done on different kinds of IF regimes, which models they used and effects, both positive and adverse. This will help the readers immensely.
3. There should be a glossary of terms, some terms have not been explained and are very technical and need to be described.
4. The review needs some language editing. Few sentences need radical corrections e.g., The first few lines of the section 2 (Effects of IF on metabolic complications): “ contributing to around 62% per 100,00 individuals”.
Comments on the Quality of English LanguageMinor edits are required, as mentioned to the authors.
Round 2
Reviewer 1 Report
Comments and Suggestions for Authors
The modified version is acceptable.